# Ground-Based Radar Interferometry for Monitoring the Dynamic Performance of a Multitrack Steel Truss High-Speed Railway Bridge

**Qihuan Huang [1,\*], Yian Wang [1,2], Guido Luzi [3], Michele Crosetto [3], Oriol Monserrat [3], Jianfeng Jiang [1], Hanwei Zhao [4] and Youliang Ding [4]**

1   School of Earth Sciences and Engineering, Hohai University, JiangNing District, Nanjing 211100, China;
    wangyian@whu.edu.cn (Y.W.); jfjiang@hhu.edu.cn (J.J.)
2   School of Remote Sensing and Information Engineering, Wuhan University, Wuhan 430079, China
3   Centre Tecnològic de Telecomunicacions de Catalunya (CTTC), Geomatics Division,
    08000 Castelldefels, Spain; gluzi@cttc.cat (G.L.); mcrosetto@cttc.cat (M.C.); omonserrat@cttc.cat (O.M.)
4   Key Laboratory of C&PC Structures of the Ministry of Education, Southeast University,
    Nanjing 210096, China; civil_hwzhao@seu.edu.cn (H.Z.); civilding@seu.edu.cn (Y.D.)
\*   Correspondence: InSAR@hhu.edu.cn

**Abstract:** With the continuous expansion of the high-speed railway network in China, long-span railway bridges carrying multiple tracks demand reliable and fast testing procedures and techniques. Bridge dynamic behavior analysis is a critical process in ensuring safe operation of structures. In this study, we present some experimental results of the vibration monitoring of a four-track high-speed railway bridge with a metro–track on each side: the Nanjing–Dashengguan high-speed railway bridge (NDHRB). The results were obtained using a terrestrial microwave radar interferometer named IBIS-S. The radar measurements were interpreted with the support of lidar point clouds. The results of the bridge dynamic response under different loading conditions, including high-speed trains, metro and wind were compared with the existing bridge structure health monitoring (SHM) system, underlining the high spatial (0.5 m) and temporal resolutions (50 Hz–200 Hz) of this technique for railway bridge dynamic monitoring. The detailed results can help engineers capturing the maximum train-induced bridge displacement. The bridge was also monitored by the radar from a lateral position with respect to the bridge longitudinal direction. This allowed us to have a more exhaustive description of the bridge dynamic behavior. The different effects induced by the passage of trains through different tracks and directions were distinguished. In addition, the space deformation map of the wide bridge deck under the eccentric load of trains, especially along the lateral direction (30 m), can help evaluating the running stability of high-speed trains.

**Keywords:** multitrack railway bridge; dynamics; displacements; microwave radar interferometer

## 1. Introduction

High-speed trains produce strong dynamic effects on the bridge girders; and, conversely, the running stability of high-speed trains poses new requirements for the girder deformation of operating bridges [1]. With the continuous expansion of the high-speed railway network in China, a large number of bridges carrying multiple railway tracks are being put into operation. Monitoring bridge displacements is a difficult task for the bridge SHM systems [2]. Displacements of railway bridges and subsidence in the transition zones have been monitored using satellite SAR interferometry [3–5]. The dynamic impact of trains on high-speed railway bridges is an important problem in the design and evaluation of such bridges. High-speed trains excite the vibrations of the bridge structures, which can

affect the performance and service-life of bridges. Moreover, the bridge vibration in turn greatly affects the running safety and stability of high-speed trains. Therefore, the train-induced vibration of bridges must be measured and evaluated in the operation of high-speed railway bridges.

Pointwise sensors, such as total station [6], accelerometers [7] and global navigation satellite system (GNSS) [8–10] are widely used for bridge dynamic data acquisition. These sensors are reliable but need to be installed in contact with the surveyed structure and their placement is often time consuming and sometimes unpractical. Video techniques [11] have the advantage of establishing a good compromise between the acquisition frame rate and the resolution, while many factors, such as artificial lights, light refraction phenomena, variation of the natural light, fog and precipitation may affect the precision of video systems. Laser sensors can be used for remote monitoring; however, they are sensitive to dust and abrupt changes of the atmospheric conditions, often resulting inadequate for in-field application [12], and also low cost camera based image processing has been experienced, see [13].

In the past decade, a coherent microwave sensor has been investigated to monitor the vibration characteristics of civil structures; the first pioneering experiment dates back to 1999 [14]. Reviews of this technique, called ground-based radar interferometry, are provided in [15–17]. With the development of commercial systems [18,19], the dynamic response of a variety of structures, such as chimneys [20], wind farms [21], towers [22,23], urban buildings [22–24] and cables [22,23], has been described. The technique demonstrated to have a high sensitivity to displacements, which under the best measurement conditions can be down to a few microns.

The interest in the dynamic behavior of bridges using ground-based radar interferometry has increased in recent years [23,25]. Ref. [26] report a method to detect both the vertical bending and the torsional movements of a section of a bridge. In the former application, the radar sensor was put below the bridge, as far as possible from the deck. In this configuration, the radar line-of-sight (LOS) was almost parallel to the bridge longitudinal direction. Due to the 1D characteristic of the radar, the radar reflecting signals with the same LOS distance were imaged in the same elemental unit, namely the radar range-bin. Therefore, different displacement behaviors in the bridge lateral direction, e.g., displacements due to trains passing in different tracks could not be distinguished. In ([27]) the authors used the IBIS-S system to study the dynamic impact of kinematic excitations during passage of low speed (<120 km/h) trains on the corrugated steel plate culverts; the sensor was set perpendicular to the railway, while difference of dynamic performance of the two tracks were not considered during the analysis.

Radar systems using the synthetic aperture radar (SAR) technique, such as IBIS-L and GPRI [25], have the ability of acquiring 2D SAR images: they can distinguish targets in the range and azimuth directions. However, they require several seconds or even several minutes to acquire an image [25]: hence, they cannot capture the dynamic displacements. They can be only used for long-term static displacement monitoring of, e.g., glaciers [28], landslides [29,30], dams [14,31,32], etc.

In this paper, the commercial real-aperture radar (RAR) interferometric sensor IBIS-S is used for measuring dynamic response of a multitrack steel truss railway bridge in both longitudinal and lateral directions. The dynamic responses under different loading conditions, including wind, low-speed passing metro vehicles, and high-speed passing trains (up to 200 km/h) are investigated. Radar measurements are carefully analyzed with the help of lidar data, as well as considering the bridge structure properties. The research reveals displacement details of the multitrack bridge that were never highlighted before.

## 2. Ground-Based Interferometric Radar

A radar detects and ranges objects using electromagnetic waves. It transmits microwave signals and receives echoes from targets within its antenna field of view Figure 1 shows a simple scheme of the ranging performed by real aperture radar, aimed at observing a bridge deck. Different parts of the structure are associated with "radar bin" [23]. With respect to conventional radar, which is

only capable of measuring the amplitude of the received signal, an interferometric radar can measure the phase with a sensibility that is a small fraction of its wavelength: for this reason, sub-millimeter displacements can be measured.

The signal to noise ratio (SNR) is a parameter of the radar backscattering quality: the higher, the better is the achievable accuracy [33]. To separate two or more targets in the radar range direction, the distance between them must be larger than the radar range resolution, which is dictated by the bandwidth of the radar. A radar bin is an elemental space unit, containing the response of a radar sampling footprint. The amplitudes of the received radar signals in all the range bins make a range profile, in which the outstanding peaks indicate the best reflecting part of the structure at hand.

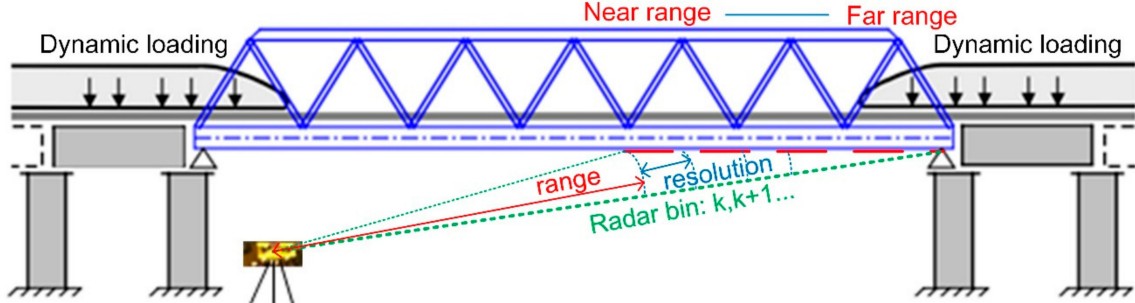

**Figure 1.** Scheme of interferometric radar for monitoring a bridge.

IBIS-S, manufactured by IDS (Ingegneria Dei Sistemi S.p.A.), is a commercial radar system with interferometric capability. The system consists of a sensor module for transmitting and receiving signals, a control PC for system controlling and data collecting, a power supply unit and data processing software. The radar sensor is mounted on a tripod equipped with a rotating head to adjust the bearing of the antenna towards the investigated structure. The sensor transmits an electromagnetic signal at a central frequency of 17.2 GHz (Ku-band) with a maximum bandwidth of 300 MHz, corresponding to a range resolution of 0.5 m. Range resolution is intended here as the minimum distance required between two targets to be separated—i.e., seen as two different subsequent bins. For details on the radar equipment and the used radar data processing please refers to [33]. The main characteristics of the sensor are summarized in Table 1.

**Table 1.** Main characteristics of the IBIS-S system.

| IBIS-S Parameters | |
| --- | --- |
| Central Frequency/wavelength | 17.1 GHz/1.75 cm |
| Maximum distance | 1000 m |
| Maximum range resolution | 0.5 m |
| Maximum sampling rate | 200 Hz |
| Nominal displacement accuracy | 0.02 mm |

## 3. Bridge Description and Experimental Setting

### 3.1. Bridge Description

The Nanjing–Dashengguan high-speed railway bridge (NDHRB) is located in the Nanjing section of the middle and lower reaches of the Yangtze River in China; it is the largest bridge with the heaviest design loading ever built in the world [2]. The main bridge consists of two continuous steel-truss arches and approach spans, with a span configuration of $108 + 192 + 2 \times 336 + 192 + 108$ m, see Figure 2. The arches feature three main trusses spaced 15 m apart in the transverse direction. The bridge girder uses an orthotropic steel plate; the railway track (ballast track) is supported by a continuous concrete trough, which is shear-resistant with the steel plate. The deck cross-section of NDHRB is shown in Figure 3. The bridge structure was built using three types of steel: Q345qD, Q370qE and Q420qE.

The bridge includes six tracks: two tracks of the Beijing–Shanghai high-speed line, two tracks of the Shanghai–Wuhan–Chengdu railway lines and two tracks on the outer sides of the bridge deck for the Nanjing Metro, see Figure 3. The designed train speed is 300 km/h, and the designed load is more than 600 kN/m in the longitudinal direction. All the lines are nowadays operative.

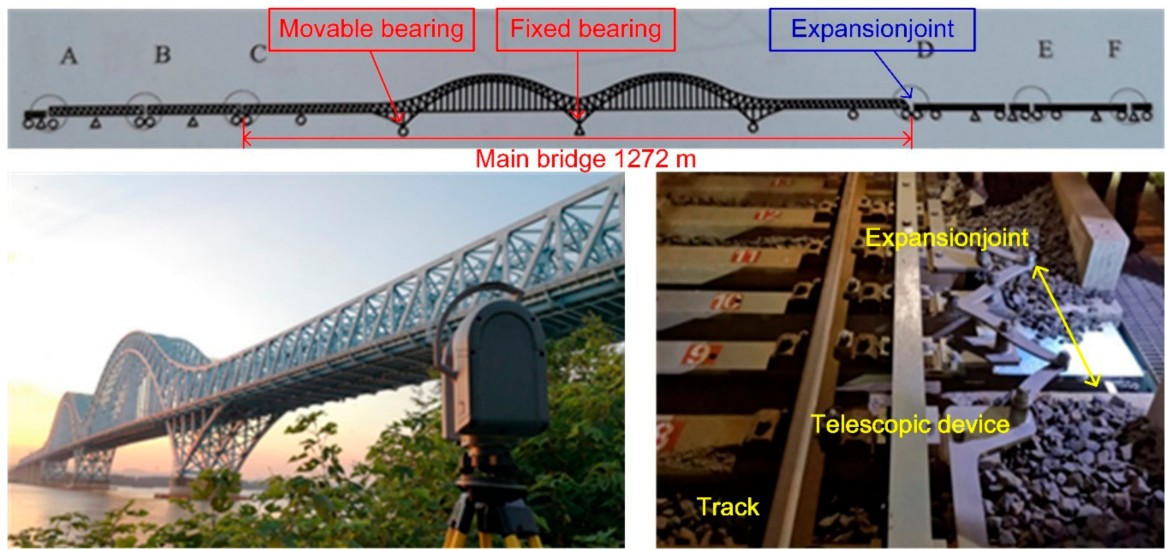

**Figure 2.** Layout (above) and view (below, left) of Nanjing–Dashengguan high-speed railway bridge (NDHRB). Below, right: telescopic device mounted at locations 'C' and 'D' shown in figure above.

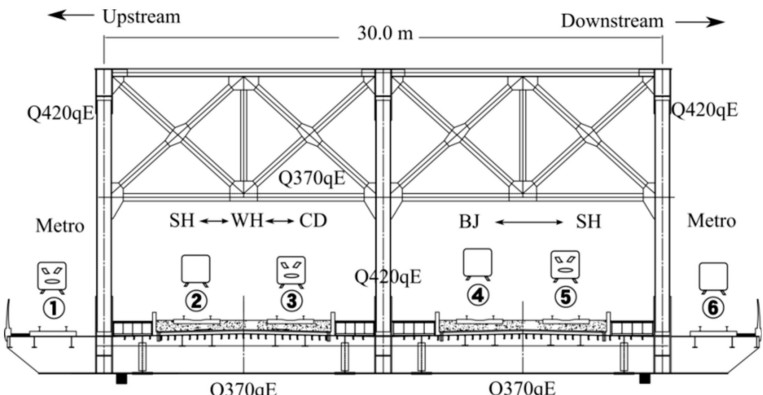

**Figure 3.** Cross-section of the NDHRB deck showing different uses of the tracks. ① Nanjing Metro Line S3 Upstream (S3-U); ② Beijing–Shanghai (BJ-SH) high-speed railway; ③ Shanghai–Beijing (SH–BJ) high-speed railway; ④ Shanghai–Wuhan–Chengdu (SH–WH–CD) high-speed railway; ⑤ Chengdu–Wuhan–Shanghai (CD–WH–SH) high-speed railway; ⑥ Nanjing Metro Line S3 downstream (S3-D).

Fixed and movable bearings were mounted on piers to support the bridge and adapt to the displacement difference between each sections. Telescopic devices were installed at each expansion joint. Due to the large longitudinal displacements at both sides of the main bridge, eight SA60-1200B telescopic devices, with displacement adaption of +/−400 mm, were installed at four high-speed railway tracks in the two sides, see Figure 2.

The bottom of the bridge exhibits a set of transversal steel beams every 3.0 m, which fit together with the three main steel trusses in the longitudinal direction and support all the corrugated steel plates. Thus, from the RAR viewpoint, the bridge appears as a composition of several high reflecting corners and edges along its entire bottom surface, which is good for radar reflection.

Due to the remarkable characteristics of NDHRB, including long span of the main girder, heavy design loading and high speed of the trains, a long-term structure health monitoring (SHM) system was installed on the bridge [2].

### 3.2. Experimental Settings

The IBIS-S sensor was used to study the dynamic behavior of NDHRB. Due to the symmetrical span configuration of NDHRB, the half bridge in the south was taken as representative. Three positions along half part of the bridge were selected to perform the IBIS-S measurements (S1, S2, S3). Four views (view S1, view S2a, view S2b, view S3), both in longitudinal and lateral direction, were captured during the data collection, see Figure 4. The positions of two hydrostatic leveling sensors corresponding to the middle of 336-m span and 192-m span are highlighted in this figure.

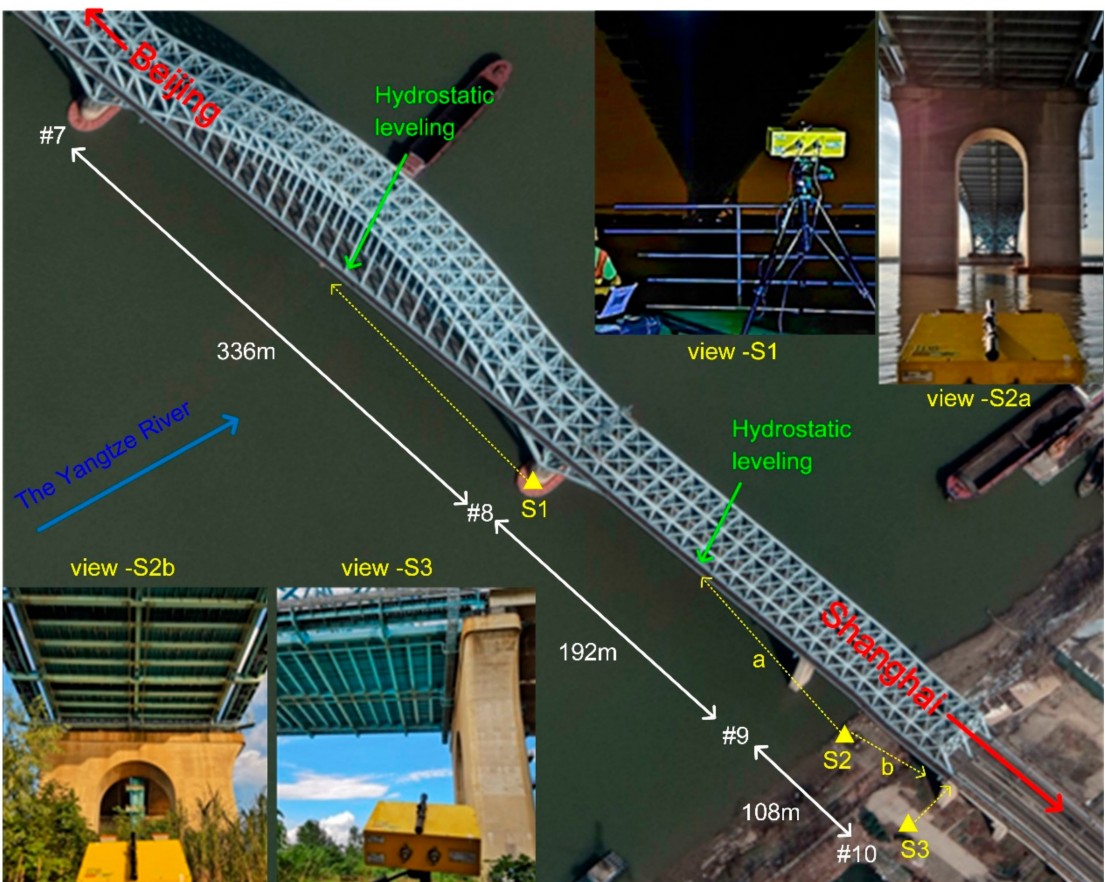

**Figure 4.** Google Earth image of the NDHRB southern part. The IBIS-S installations (S1, S2, S3) are marked with yellow triangles and the approximated radar LOS is highlighted with yellow dotted lines; the four pictures show the corresponding views (view S1, view S2a, view S2b and view S3); the positions of the hydrostatic leveling are also indicated with green arrows.

## 4. Results of the Bridge Dynamic Responses

### 4.1. Ambient Vibration of the 336-m Span

S1 was set on the basement of the pier #8, at the center of the bridge width, and the IBIS-S sensor was located 28.6 m below the bridge. The radar antennas were pointed to the middle of the 336-m span. The field campaign was carried out during the bridge maintenance (00:30 to 04:30, 10th July 2018) and without train activity. This field campaign aimed to study the ambient vibration behavior of the 336-m span. The radar instrument was working in the dynamic mode with a maximum range

of 250 m. The total duration of the acquisition was approximately ten minutes, at a sampling rate of 148.91 Hz and 2.0-m of resolution in range.

Figure 5 shows the range profile of the bridge deck. Due to the corner reflector effect of the steel beams in the lateral direction, a fairly high SNR (>55 dB) was obtained for all the bins. The radar bin corresponding to the middle of the span, i.e., the most sensitive position, was selected as representative to obtain the vibrations. A portion of the time history of this bin is depicted in Figure 6a. This was compared to the synchronized result achieved from the hydrostatic leveling installed at the same position and measuring with a sampling rate of 1 Hz (Figure 6b). To perform the comparison, all the displacements were projected along the vertical direction, which was considered the main displacement component. To the noted, the hydrostatic leveling has a relative lower precision (±0.1 mm) compared with the radar IBIS-S. On contrary, the IBIS-S has a displacement accuracy of 0.02 mm, while it can be affected by the change of environment. We can see from Figure 6 that: the magnitudes of the displacements are both within ±0.3 mm and non-periodic displacement behavior was observed from both sensors. The inconsistency in figure can be explained as the uncertainty of the measurements. This agrees with the rigid characteristic of a railway bridge, which has a relatively rigid response to ambient noise.

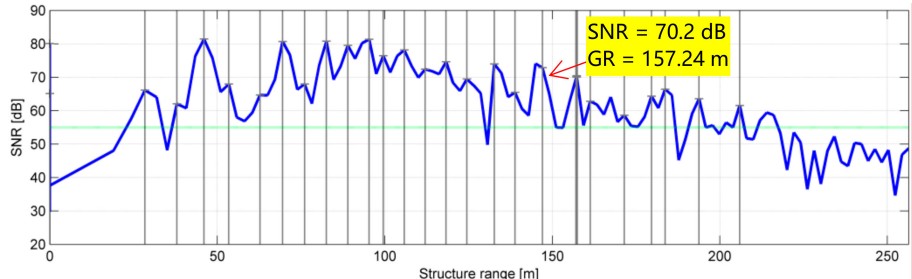

**Figure 5.** Range profile of the bridge deck. Bold gray line indicates the radar bin (number 76) selected for estimating the deck displacements.

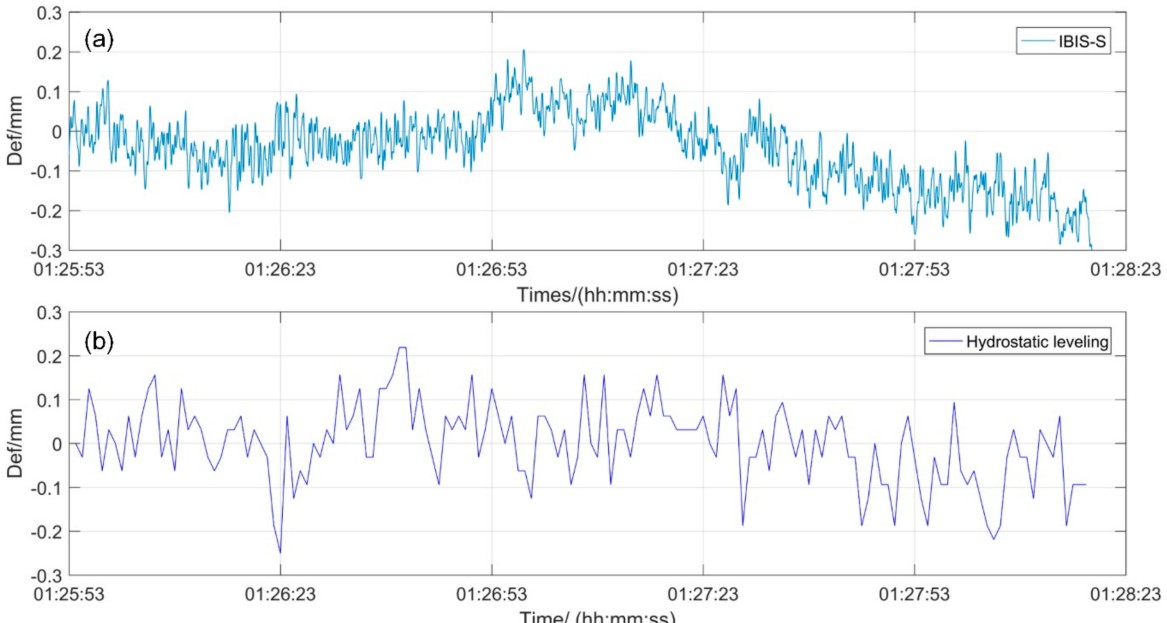

**Figure 6.** Displacements in mm due to ambient vibration in the middle of the 336 m-span achieved from (**a**) IBIS-S and (**b**) the hydrostatic leveling.

### 4.2. Dynamic Vibration of the 192-m Span

To monitor and evaluate the dynamic behavior of NDHRB, an ideal configuration would be to set the IBIS-S system at pier #7 or #8, aiming at the middle of the 336-m span. However, the bridge can only be accessed in the window time when there are no trains. Therefore, the Station S2 was chosen and the 192-m span was selected for the dynamic vibration monitoring.

S2 was set at the south riverside, right underneath the bridge between piers #9 and 1#0. Two views (view S2a/b) were acquired: the radar signal of view S2a was transmitted through the archway of pier #9, aiming at studying the dynamic behavior of the 192-m span between piers #8 and #9. view S2b corresponds to the radar signal from the 108-m span between piers #9 and 1#0, aiming at monitoring the behavior of the beam end, where the telescopic devices are located. view S2b is analyzed in Section 4.3.

A Trimble$^{TM}$ TX8 3D lidar—with maximum operating range of 340 m—was used for collecting the point clouds and supporting the geometric analysis of the radar measurements. The distances from S2 to pier #9 and 1#0 are 61 m and 47 m, respectively and the height to the bridge deck is 35 m.

The view S2a observation geometry and the corresponding radar range profile are illustrated in Figure 7. In the view S2a campaign, the radar worked in the range 70–250 m. The sampling rate was 53.85 Hz, with 2.0-m resolution in range. The data acquisition started at 18:04, on 13th July 2018 and lasted 26 min. Different loading cases were observed during the campaign. We selected six loading conditions described in Table 2 as representatives for the study of the bridge dynamic behavior.

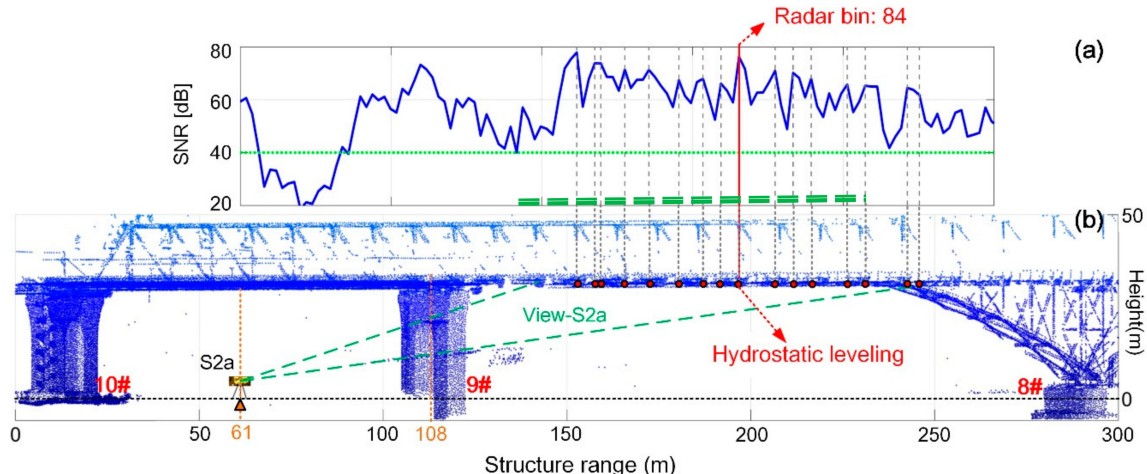

**Figure 7.** (**a**) Range profile of view S2a: the vertical dotted gray lines indicate the 16 selected radar bins (number 57, 60, 61, 65, 69, 74, 78, 81, 84, 90, 93, 96, 102, 105, 112 and 114) for estimating the deck displacement; (**b**) scheme of the radar geometry using the lidar point cloud. The red line indicates the radar bin (number 84) and the corresponding position of the hydrostatic leveling, in the middle of the 192-m span.

**Table 2.** Loading cases of view S2a.

| Cases | Track No. | Direction | Carriages | Speed (km/h) |
|-------|-----------|-----------|-----------|--------------|
| **Case1** | ③ | N2S | 16 | 237.7 |
| **Case2** | ④ | S2N | 16 | 244.2 |
| **Case3** | ③&⑤ | N2S / N2S | 16 / 16 | 244.3 |
| **Case4** | ②&⑤ | S2N / N2S | 16 / 16 | 248.1 |
| **Case5** | ⑥ | S2N | Metro | ~80 |
| **Case6** | ①&④ | N2S / S2N | Metro / 16 | 213.1 |

The measured LOS displacements, $d_{LOS}$, can be composed in a vertical displacement $d_V$ and an horizontal displacement $d_H$, while the latter one can be decomposed into the longitudinal displacement $d_L$ and the lateral displacement $d_{Lat}$. The S2a acquisition geometry is almost perpendicular to the

lateral direction: this results in a negligible sensitivity to the lateral displacements. Considering the measurement geometry of view S2a, it is more sensitive to the longitudinal displacements than to the vertical ones. However, the longitudinal displacements are only present in some particular cases, such as nonuniform motions of the trains, irregularities at the wheel-to-rail interface and the centrifugal load on the curved movement path. Figure 8 shows the simultaneous displacement time series of the horizontal components measured by the bridge SHM system during the view S2a monitoring campaign. It can be seen that the lateral displacements (Figure 8a) in the middle of the 192-m span are within ±0.1 mm, and the dynamic response of passing trains can be identified by the larger value magnitude. Variation of the longitudinal displacements at pier #8 (Figure 8b) are approximately 4 mm. It can be seen that the lateral displacement (less than 0.1 mm) is mainly affected by the train, while the longitudinal displacement (4 mm) is mainly affected by the ambient temperature variation (around 1 °C), which agrees with the calculated thermal expansion (4 mm) of the steel truss (coefficient of thermal expansion: $13.0 \times 10^{-6}/°C$, length to the fixed pier #7: 336 m). No clear relation between the longitudinal displacements and passing trains was found. Thus, in the following, we neglect the horizontal displacement components, and the deformation is considered vertical. It is worth noting that the trend of the radar-measured displacements can be associated with a lowering of the of air temperature, which affected the radar phase measurement.

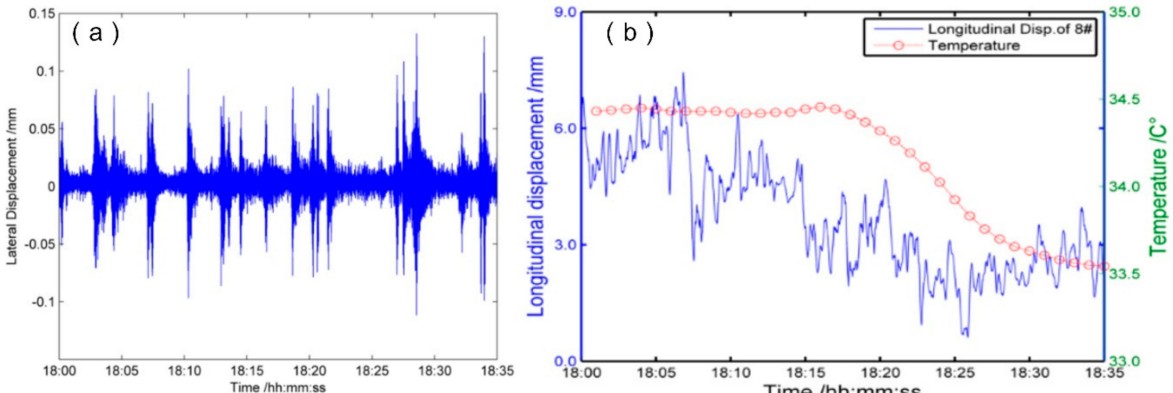

**Figure 8.** Time series of the horizontal displacements. (**a**) Lateral displacement in the middle of the 192-m span; (**b**) longitudinal displacement and temperature at pier #8 from the existing structure health monitoring (SHM) system.

Figure 9 shows the obtained vertical displacement time series in the middle of the 192-m span. This result was compared to the synchronized hydrostatic leveling measurements at the same position. We summarize below the main results:

(1)    As far as the train-induced displacement procedures are considered, the radar and hydrostatic leveling results are quite consistent in the six different load cases;

(2)    The hydrostatic leveling results about the train-induced displacements miss the real peak in most instances. By contrast, the radar results provide a higher detail in the description of the displacement behavior. This is due to the higher sampling frequency (53.845 Hz) with respect to leveling sensor (1 Hz). This can help the engineers to capture the maximum value of the train-induced displacements of bridge;

(3)    The displacements occur as soon as the given train reaches the main bridge, and they decrease to the normal state as far as the loading disappears. This is evident by calculating the duration of the loading on the main bridge. For instance, in Case 1 the displacement lasts for about 25 s. Considering the velocity of the train (237.7 km/h), the main bridge length (1272 m) and the train length (400 m, 16 carriages of 25 m each), the total time for the train passing the bridge is about 25.3 s. The same result can be achieved for the metro, considering the 6 carriages of 20 m each, with a speed of about 80 km/h;

(4)  The temporal evolution of the displacements is related to the direction of the passing train. When the train arrives, small displacements are detected at the beginning; then they transmit like a sine wave, while the displacement amplitude becomes larger and larger; when the train leaves the bridge, the displacements disappear quickly. The behavior is symmetric for the trains coming from north to south (N2S) and from south to north (S2N);

(5)  The vertical displacements achieve the maximum value when the train is running on the measuring point. In all cases, the displacement magnitude is less than 12 mm;

(6)  The magnitude of vertical displacement, as well as the duration, does not significantly increase when two trains are running simultaneously on the bridge. This is in agreement with the structural characteristic of NDHRB, which is a rigid bridge.

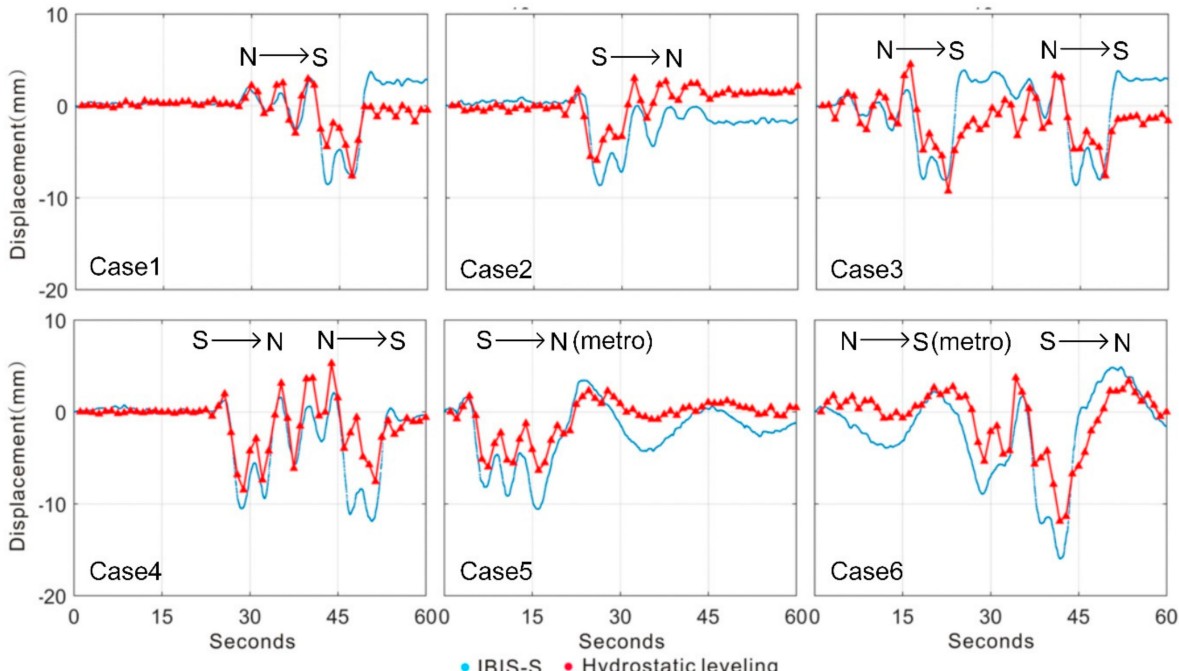

**Figure 9.** Vertical displacement time histories in the middle of 192-m span monitored by IBIS-S (blue lines) and the hydrostatic leveling system (red triangles).

Bins corresponding to the bridge deck were analyzed to detect the main frequencies of the displacement measured by the radar. In Figure 10 we show the displacement retrieved for bin 83, bin 84 and 85. The power spectral density (PSD) calculated using the periodogram procedure with the Welch method [34], after filtering the time sample using a Butterworth bandpass filter of third order whose bandpass (0.3 Hz–3 Hz) is shown for bin 83 and bin 84 in Figure 11. The goal of using this filter is to emphasize the stationary response with respect to the pulse response due to the train passage. The entire sample was divided into two, partially overlapping, signals to also evaluate the stationarity of the acquisition. Observing Figure 11 we can clearly find out some peaks at frequencies close to that measured by the accelerometers (Figure 12), in which the data were filtered (0–20 Hz) in frequency domain. Table 3 report the main value compared to those from the accelerograms. Observing the values of table we can make some comments: The lowest (0.267 Hz) clearly appears only for bin 83, so we can discard it as a vibration not of the main structure of the bridge; The two set of data agree within less than 10% of error between the two techniques results, although radar results are noisier; According to the frequencies obtained by the radar, the main/similar frequencies of the bridge were recognized by the radar, hence the results are credible.

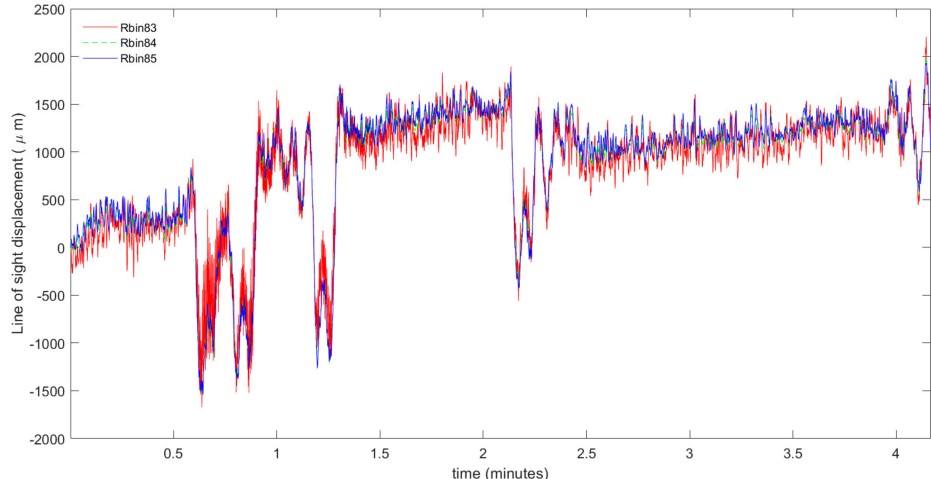

**Figure 10.** Line-of-sight (LOS) displacement time histories in the middle of 192-m span monitored by IBIS-S.

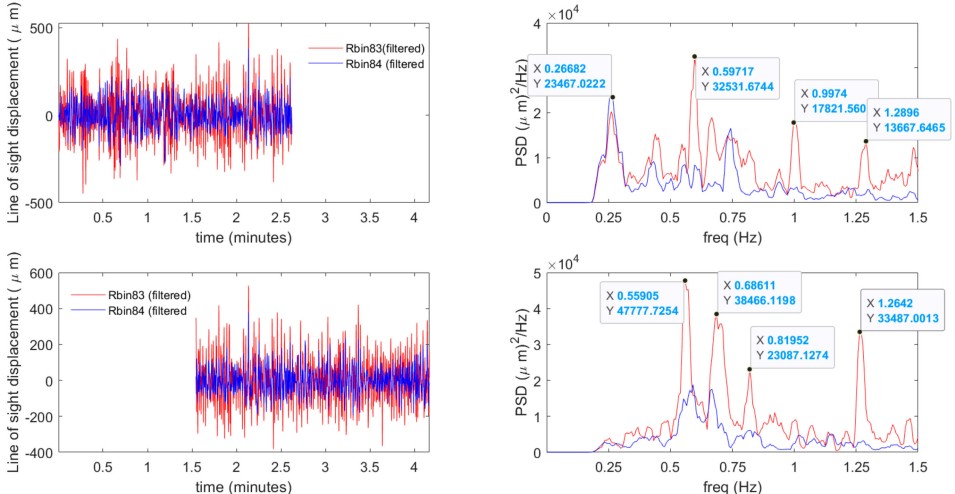

**Figure 11.** Filtered LOS displacement time histories of bin83 and bin84 with corresponding periodograms spectra calculated for two sub-samples with a partial overlap.

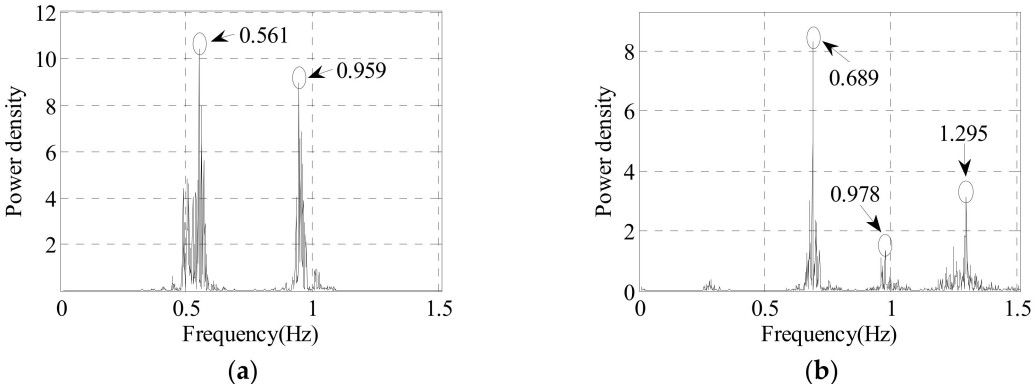

**Figure 12.** Transverse and vertical frequencies measured by accelerometers. (**a**) Transverse acceleration (**b**) vertical acceleration.

**Table 3.** Frequency values obtained from radar data, compared to accelerometer results.

| Accelerometer | | RADAR |
|---|---|---|
| **Transverse/Hz** | **Vertical/Hz** | **LOS Displacement/Hz** |
| | | 0.267 |
| 0.561 | | 0.597/0.599 |
| | 0.689 | 0.686 |
| | | 0.820 |
| 0.959 | | |
| | 0.978 | 0.997 |
| | 1.295 | 1.264/1.290 |

Figure 13 illustrates the vertical displacement maps of the bridge deck using the displacement history of 16 radar bins, which are almost uniformly distributed in the bridge deck. Due to the obstacle of the pier #9, only part of the span was observed (between 70 m and 170 m to the pier #9, see Figure 7). It can be seen that the maximum displacement was observed near the middle part of the span (marked with dashed line in each cases) and the displacement are slightly different from the near range and the far range, depending on the direction of the moving train.

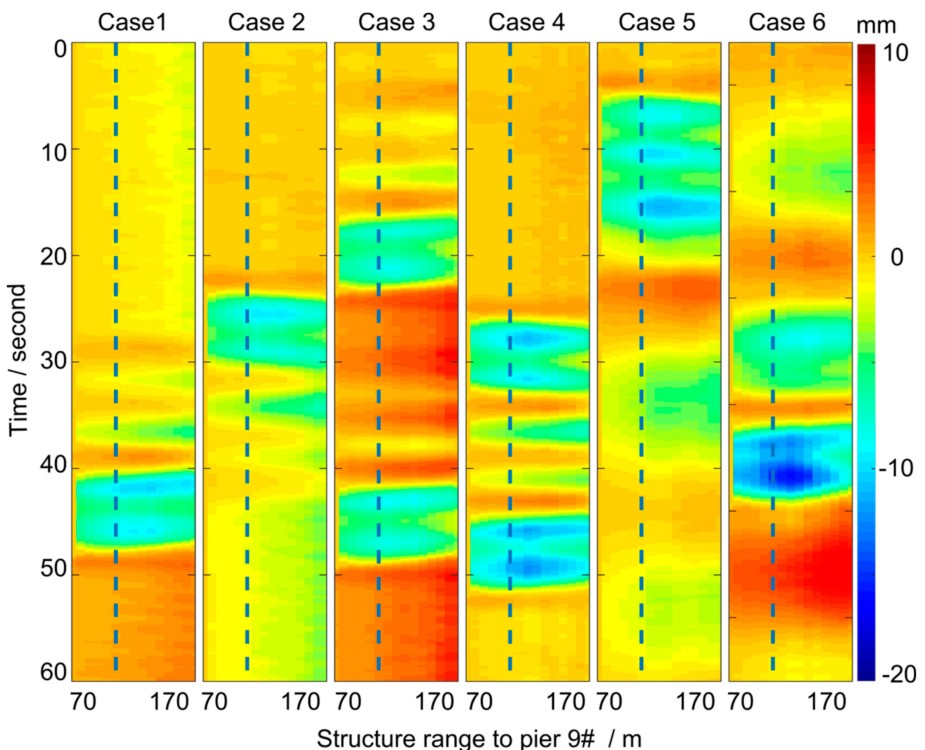

**Figure 13.** Vertical temporal displacement maps of the 192-m span for the analyzed six cases. X axis (70 m~170 m) shows the distance to pier #9; dashed lines indicate the middle of the 192-m span.

*4.3. Dynamic Behavior at the Expansion Joints*

Two perpendicular radar views, i.e., view S2b, in the longitudinal direction and view S3, in the lateral direction, were set to monitor the bridge dynamic behavior at the expansion joint.

4.3.1. View-S2b and Results

View S2b observed the expansion joint from the longitudinal direction right underneath the bridge. The radar was set in the dynamic mode with a maximum distance of 100 m. The sampling rate was 200.19 Hz, with a 2.0-m resolution in the range. The data collection started at 18:33 on 13 July 2018

and lasted more than 20 min. Figure 14 shows the view S2b geometry and a range profile of the radar backscattering. A very high SNR (>65 dB) was measured due to the corner reflector effect of the steel beams and a relative short range. Different loading cases were observed during the radar acquisition. We selected the four loading cases described in Table 4 as representative ones.

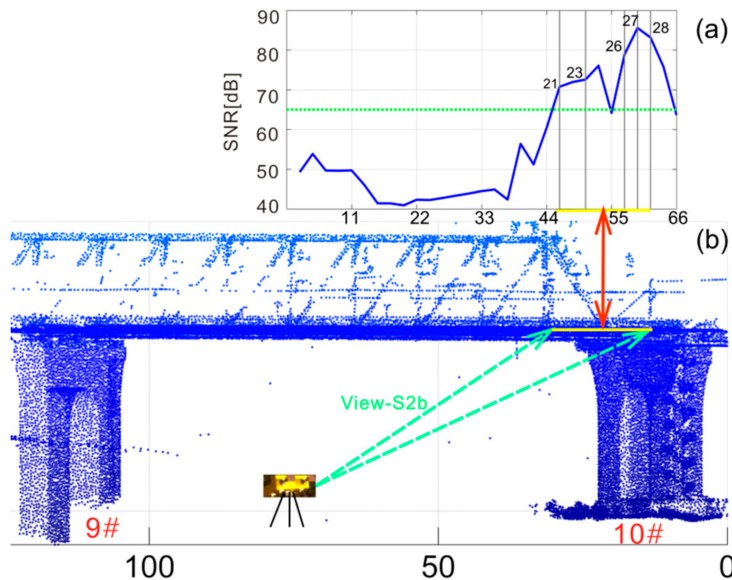

**Figure 14.** (**a**) Range profile of view S2b. Vertical gray lines indicate the radar bins 21, 23, 26, 27 and 28; (**b**) scheme of the radar geometry using the lidar point clouds.

**Table 4.** Loading cases of view S2b.

| Cases | Track No. | Direction | Carriages | Speed (km/h) |
|---|---|---|---|---|
| 1 | ② | S2N | 8 | 246.3 |
| 2 | ③ | N2S | 16 | 243.3 |
| 3 | ⑥/⑤ | S2N/N2S | Metro/16 | ~80/246.8 |
| 4 | ⑤/① | N2S/N2S | 8/Metro | 244.8/~80 |

Figure 15 shows the vertical displacement time histories of the radar bin 21, 23, 26, 27 and 28. The displacement histories include two parts: the first part is mainly caused by the loading of passing train and the second one is its aftershock. The shape of the displacement history is related to the direction of passing train, which has the same characteristics as those at view S2a. The duration of the first part depends on the length of the train and its speed: it takes approximately three seconds for a train with 8 carriages and six seconds for a train with 16 carriages. This agrees with a carriage length of 25 m. The duration of the second part is more than 15 s for both 8- and 16-carriage trains. The maximum vertical displacement of the second part is less than 1 mm for all the cases, while the 16-carriage train induced larger vertical displacement (4.2 mm) in the first part one respect to the 8-carriage train.

In each case, the differences of the vertical displacement amplitudes of the measured bins in the second part are very small, while those in the first part have a clear relation with the position of the bins: the farer the distance to the pier 1#0, the larger the displacements.

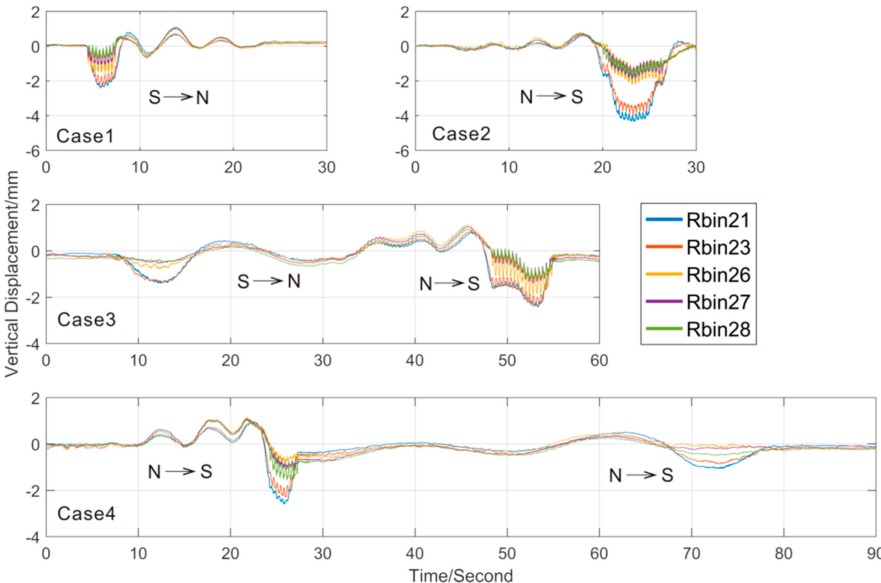

**Figure 15.** Vertical displacement time histories of the four loading cases.

### 4.3.2. View S3 and Results

S3 was set perpendicular to the bridge longitudinal direction. This special setting provides high sensitivity to dynamic performance of the bridge deck in its lateral direction. This allowed distinguishing the vibration behavior due to trains running on different tracks and how the vibration affects the telescopic device at the expansion joints. The S3 dynamic mode sampled at a rate of 166.39 Hz, with 0.5-m resolution in the range, over a maximum measurement distance of 100 m.

Figure 16 depicts the radar reflected signal and the bins selected for bridge displacement monitoring. Due to the short LOS distance (less than 80 m) and the corner reflector effect of the steel truss, the bridge deck has a very strong backscattering: the SNR of the radar signal scattered from the bridge deck is more than 60 dB. Eleven radar bins, excluding the two bins at the two sides of the structure (corresponding to the metro tracks, whose SNRs are less than 55 dB), were selected to estimate the dynamic behavior induced by high-speed train, running on all the different tracks, see Table 5.

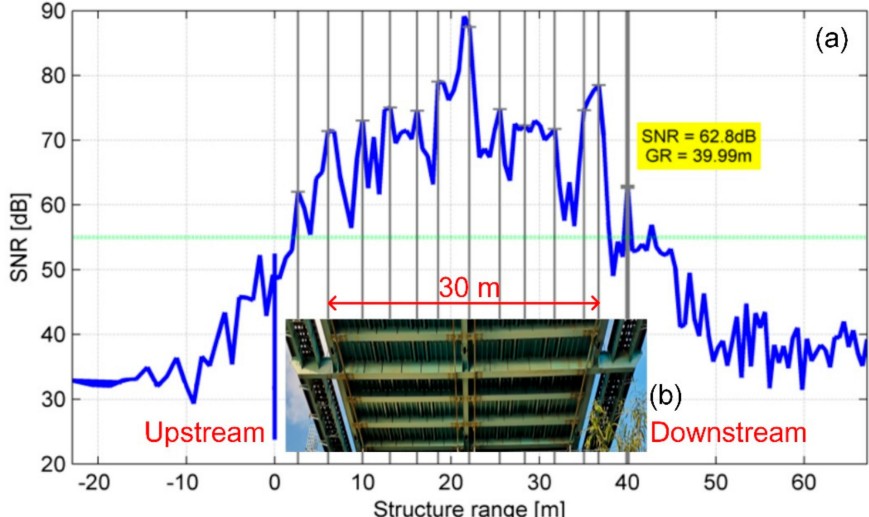

**Figure 16.** (**a**) Structure range profile of the bridge deck and the gray lines indicate the radar bins (numbers 87, 93, 98, 103, 108, 113, 119, 124, 127, 135, 142, 144 and 150) selected for estimating the deck displacement; (**b**) photograph from the view S3 in Figure 4, which shows the bottom of the bridge deck illuminated by the radar.

**Table 5.** Loading cases of view S3.

| Cases | Track No. | Direction | Carriages | Speed (km/h) |
|-------|-----------|-----------|-----------|--------------|
| **C1** | ④ | S2N | 16 | 243.5 |
| **C2** | ③ | N2S | 16 | 197.4 |
| **C3** | ⑤ | N2S | 16 | 247.1 |
| **C4** | ③ | N2S | 16 | 247.5 |
| **C5** | ③&④ | N2S/S2N | 8/16 | 245.1/245.1 |
| **C6** | ③ | N2S | 16 | 220.3 |

Figure 17 depicts the vertical displacement map of the bridge deck which is near the expansion joint. The displacement induced by high-speed train mainly affects the swath where it is running. This phenomenon is clearer when the train is running on track 4 and track 5, e.g., in Case 1, Case 3 and the second part of Case 5. By contrast, in Case 2, Case 4, Case 6 and the first part of Case 5, where the trains were running on the track 3, there are also induced displacements in the other swaths. This phenomenon does not depend on the direction of the train and the load of the train. Therefore, it is necessary to inspect the structure of lane 3 to determine whether it affects the safety of the train. The vertical displacement map shows the space deformation of the bridge deck, especially along the lateral direction (30 m), which controls the running stability of high-speed trains.

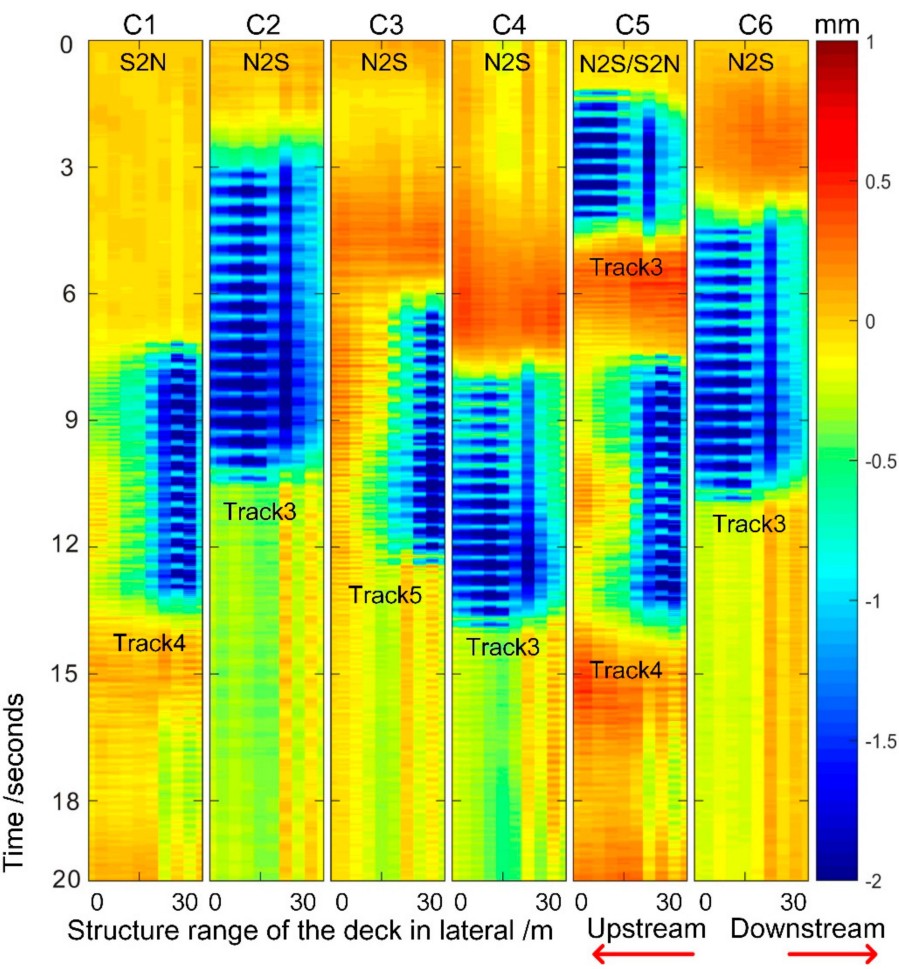

**Figure 17.** Vertical displacement map of the bridge deck near the expansion joint.

The vertical displacement time histories of the bridge deck at upstream and downstream swaths are compared in Figure 18. Although there is some slight difference between the displacement responses obtained from the six dynamic loading cases, the nature of the response curves is very

similar. The displacement is mainly caused by the train loading, and the duration of the displacement is related to the length of the train and its speed, as already commented for view S2b. It can be seen that the amplitude of the vibration is approximately 2 mm to the swath which has train loading, while the magnitude of the swath without loading is much smaller. This can be explained considering that the three main trusses have the effect of separating vibration induced by high-speed passing train.

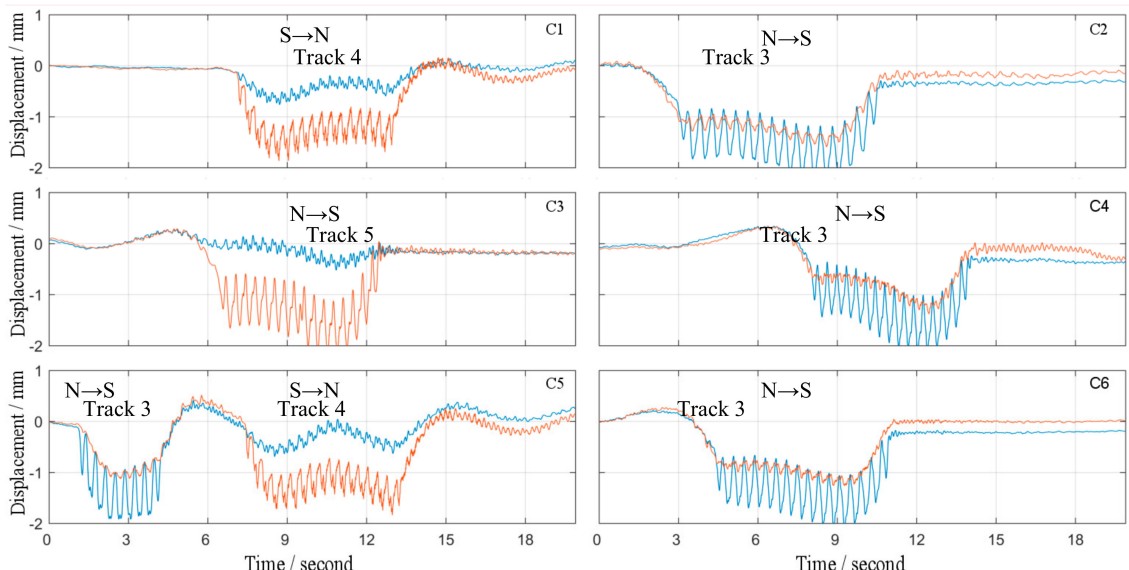

**Figure 18.** Comparison of vertical displacement histories of the bridge deck at (light blue) upstream and (orange) downstream swath.

## 5. Discussion

One of the main limitations of the use of RAR measurements for bridge monitoring is the capability to only estimate the LOS of the displacement. This limits the possibility of obtaining a full picture of the bridge vibrations. Although the vertical displacements are usually the most significant component, the horizontal displacements (longitudinal and lateral) cannot be neglected. A possible approach to overcome such a limitation is to use different radar sensor measuring simultaneously from different positions: the integration of all measurements allows to decompose the measured displacement in its different components. In this work, we have used a single RAR system measuring from different positions. The in situ data provided by an existing SHM system of the bridge were used to check that both lateral and longitudinal displacements are negligible with respect the vertical ones. Hence, all the measurements were projected to the vertical direction.

The results obtained in view S1 and view S2a were compared with the results obtained using external sensors mounted on the bridge. In view S1, the dynamic responses to ambient noise are within ±0.3 mm, which, in fact, are uncertainty of the displacements measurement and the result agree with the rigid characteristic of the bridge; In view S2a, we first analyze the horizontal displacements achieved by the in situ sensors and then the displacement obtained from the RAR and the in situ measurements are compared, the results confirms the assumption that the train-induced displacements are vertical. The main difference between the in situ and the RAR measurements is the sampling frequencies, the high frequency of the radar measurements are capable of illustrating the maximum train-induced displacements.

Two measurements of the same section of the bridge with perpendicular LOS (view S2b and view S3) were performed in order to emulate the measurement of a two-RAR system. Although the two measurements were not simultaneous, we compared measurements with similar loads on the bridge. The loading cases of 16 carriages high-speed trains passing on the bridge (track ③, N2S) were chosen. Due to the fact that the train-induced lateral displacements are within 0.1 mm (see Figure 8a), Figure 19

depicts the converted displacement in the vertical direction. In Figure 19a, the three radar bins (number 26, 27 28) have exactly the same displacement behavior, while the magnitudes of vibration are different. This is because the bins have different distances from the pier 1#0: the farer the distance, the larger the magnitude. In Figure 16b, the displacements have a similar behavior of that shown in Figure 16a. The largest displacements are related to the part of the eccentric train loading of the bridge: the farer to the eccentric loading, the smaller the displacement.

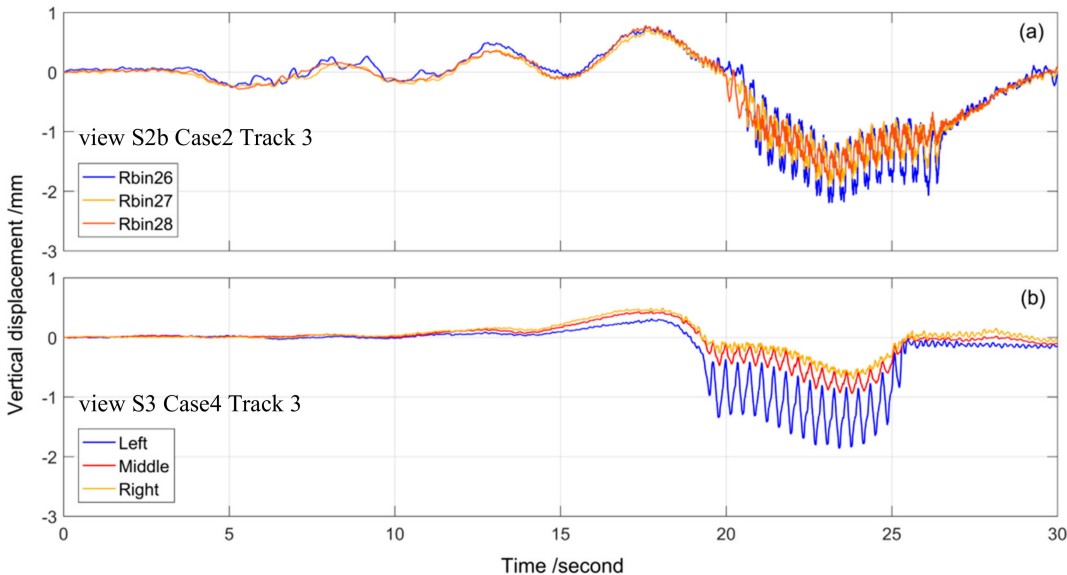

**Figure 19.** Vertical displacements in (**a**) view s2b case 2 track 3 and (**b**) view s3 case 4 track 3. the 'left', 'middle' and 'right' are the radar bins selected at the corresponding position in the bridge lateral.

Due to the fact that the train-induced displacements are in a 2D plane of the bridge deck, while the two radar LOS displacements are on the two perpendicular lines in the plane, a quantitative analysis of the two measurements becomes complicated.

The RAR can only measure in the LOS and cannot measure the displacements perpendicular to the LOS. Usually, the displacements in the bridge lateral direction can be considered uniform. However, if a bridge is quite wide, e.g., including multiple tracks/lanes, the displacement can be laterally nonuniform. Setting a measurement perpendicular to the bridge longitudinal direction allows monitoring such variations.

## 6. Conclusions

This paper describes some experiments aimed at monitoring the dynamics of a multiple-track railway bridge (NDHRB) using a microwave radar interferometer named IBIS-S. Three stations (S1, S2 and S3) with four radar views (view S1, view S2a, view S2b and view S3) were set to monitor different parts of the bridge dynamic and monitor the dynamic responses under different loading cases: ambient noise and wind, high-speed train passages and metro passages. The radar was positioned underneath the bridge, looking the bridge longitudinally and perpendicularly to the bridge. Lidar point clouds measured by Trimble[TM] TX8 3D scanner were used for interpretation. The RAR data were compared with those coming from the bridge SHM system. The RAR data showed better results in describing the dynamic behavior of the bridge: the dynamic details and the maximum of bridge displacement can be properly captured using the RAR data. In addition, the spatial features of the bridge deck deformation under the eccentric train loads was properly described using the RAR system. These advantages indicate that microwave interferometry is a useful technique for monitoring the dynamics of bridges with multiple tracks.

Here below are summarized the main conclusions related to the experimental results of this work:

(1) The radar measurements at S1 show that, for a 336-m steel truss span, the magnitude of ambient vibration in the bridge vertical is within ±0.3 mm. This response agrees with the rigid characteristic of the railway bridge at hand;

(2) The experimental results at S2 show that, for a 192-m steel truss span, the vertical displacements induced by high speed passing train (250 km/h) are less than 12 mm; the pattern of the displacement time history is related to the track of train and its duration depends on the period during which the train is passing on the main bridge. Good consistency in all the representative loading cases was achieved with the in-situ SHM system data. A more detailed displacement behavior is achieved from IBIS-S due to its higher sampling frequency and the maximum displacements are captured;

(3) The lateral behavior of the bridge can be studied when setting the radar perpendicularly to the bridge, while the traditional way of sensor setting (parallel to the bridge) cannot. Measurements of the bridge deck perpendicularly to the bridge (S3) show that, for a multitrack steel truss bridge, the vertical displacements are not uniform in bridge lateral direction larger displacements are monitored in the track where the train is passing, while minor effects are present on other tracks.

**Author Contributions:** Q.H. conceived and designed the experiments; Y.W., Y.D. help to analyze the data; Q.H., Y.W. and J.J. contributed the field campaign; Q.H. prepared the manuscript, G.L., M.C., O.M. and H.Z. revised the manuscript. All authors have read and agreed to the published version of the manuscript.

**Funding:** This work is supported by the Fundamental Research Funds for the Central Universities (2018B18814, 2018B699X14) and the Postgraduate Research & Practice Innovation Program of Jiangsu Province (KYCX18_0619). The CTTC activities were partially funded by the Spanish Ministry of Economy and Competitiveness through the DEMOS project "Deformation monitoring using Sentinel-1 data" (Ref: CGL2017-83704-P).

**Acknowledgments:** We thank Yunqing Bai, from Beijing MAG Tianhong Science & Technology Development Co., Ltd., for collecting the Lidar data and Kang Yang from Southeast University, Xuyong Ying, from JSTI, for their support of the ground base radar campaign. The hydrostatic leveling data were provided by China Railway Major Bridge (Nanjing) Bridge and Tunnel Inspect & Retrofit Co., LTD.

**Conflicts of Interest:** The authors declare no conflicts of interest.

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
