# Peer review of "Ground-Based Radar Interferometry for Monitoring the Dynamic Performance of a Multitrack Steel Truss High-Speed Railway Bridge"

_remotesensing, doi:10.3390/rs12162594_

Round 1
Reviewer 1 Report
The article is well and clearly written, but may be too long.
PSD abbreviation is not described. The IBIS
and GPRI instruments should be at least briefly described in the text so that the reader does not need to refer to other articles.
Reviewer 2 Report
The paper illustrates what results can be achieved using the GB-RAR method with the one commercial real-aperture-radar (RAR) interferometric sensor IBIS-S to determine the dynamic response of a multi-track steel truss railway bridge. The different parts of the bridge and the dynamic responses under different loading cases were monitored.
Some remarks:
lines 94-96: the statement "In this paper, the commercial real-aperture-radar (RAR) interferometric sensor IBIS-S is used for measuring dynamic response of a multi-track steel truss railway bridge in both longitudinal and lateral directions." is not true. In this paper the dynamic response in both longitudinal and lateral directions measured only by the bridge SHM system are presented. All presented displacements measured (and recalculated) by IBIS-S are in the vertical (or LOS) direction – see lines 285-287. This sentence should be deleted.
Table 1: the "Nominal displacement accuracy" value will be better to give in mm (0.02 mm).
line 203: (Zhao et al., 2019) have to be given in [1] form.
lines 433-434: Figure 16. (b) is not stated. If part of Figure 4 is to be used, this must be stated - (e.g. see part of Figure 4.).
lines 440-443: This phenomenon probably does not depend on the direction of the train (S2N or N2S), but on the Track No. (part of the bridge deck) used. See Case 1 and Case 3, which have the same phenomenon but different train directions. This must also be corrected in the conclusions (line 536). For greater clarity, I recommend adding the numbers of the tracks in circles to Figures 9, 13, 15, 17, 18 and 19.
lines 494 and 498: the No. of the Figure is not 16a and 16b but 19a and 19b.
line 505: instead (c) have to be (b). For greater clarity, I recommend adding No. of the Case and Track to the Figure 19., e.g.: “… in (a) View-S2b Case 2 Track No. 3 and (b) View-S3 Case 4 Track No. 3.”
Reviewer 3 Report
The paper show an interesting application of RAR Ground Based interferometry, that shows the possibility of this technique to recover very detailed information on the behaviour of steel bridges under dynamic loads with a very high resolution for plastic and elastic deformation.
Appreciable is it is the scientific rigor in the illustration of the experimental setup and in the approach in data analysis
Regarding the formal aspects, it should be noted that:
- at lines 494 and 498 the text is referred to figure 16 instead 19
- In Figure 5 should be enlarged the yellow label referred to the curve, because it is not very readable
